# Direct and Indirect Effect via Endophytism of Entomopathogenic Fungi on the Fitness of *Myzus persicae* and Its Ability to Spread PLRV on Tobacco

**DOI:** 10.3390/insects12020089

**Published:** 2021-01-21

**Authors:** Junior Corneille Fingu-Mabola, Thomas Bawin, Frédéric Francis

**Affiliations:** 1Entomologie Fonctionnelle et Évolutive, Terra, Gembloux Agro-Bio Tech, Liège-Université, Passage des Déportés 2, 5030 Gembloux, Belgium; frederic.francis@uliege.be; 2Department of Arctic and Marine Biology, UiT The Arctic University of Norway, Framstredet 39, 2019 Tromsø, Norway; thomas.bawin@uit.no

**Keywords:** entomopathogenic fungi, endophytic colonization, insect–plant–microbe interactions, multitrophic interactions, aphid-borne virus

## Abstract

**Simple Summary:**

Aphids are major crop pests that are feeding on plant sap and transmitting plant viruses, thus inducing high yield losses worldwide. As chemical pesticides are decreasingly used in plant protection, fungi that cause disease to insects (entomopathogenic fungi) are one of the promising alternatives. They are commonly applied by spraying plants to protect them against herbivores. When applied, some fungi penetrate and live within plant tissues, thus helping to internally protect from insect attacks and other plant diseases. The aim of our study was to assess the effects of entomopathogenic fungi (EPF) applied firstly by contact after insect direct spraying, secondly by endophytic plant inoculation, and thirdly by associated both methods assessing the green peach aphid performances. The impact of the presence of endophytic entomopathogenic fungi (EEPF) in plant tissues on virus transmission by aphids was also considered. We found that the EPF *Beauveria bassiana* killed the green peach aphid and reduced its fecundity regardless of the application method. On fungal-inoculated plants, there was also a high mortality of aphid nymphs and infection by the potato leafroll virus (PLRV) was delayed by about a week with the EEPF treatment compared to fungal-free plants. This study showed that spraying plant leaves with EPF not only has a direct insecticidal effect against insects but could also have beneficial side effects for the plant against viruses.

**Abstract:**

Aphids are major crop pests that transmit more than half of all insect-vectored plant viruses responsible for high yield losses worldwide. Entomopathogenic fungi (EPF) are biological control agents mainly used by foliar application to control herbivores, including sap-sucking pests such as aphids. Their ability to colonize plant tissues and to interact with diverse plant pathogenic microorganisms have been reported. In our study, we evaluated the effectiveness of *Beauveria bassiana* ((Balsamo-Crivelli) Vuillemin) directly applied by contact or/and indirectly via endophytism in tobacco plants (*Nicotiana tabacum* L.) against the virus vector *Myzus persicae* (Sulzer) carrying the Potato leafroll virus (PLRV) or not. We found that both contact treatment and endophytic colonization of leaves significantly increased aphid mortality and decreased the fecundity rate when compared to control plants. In addition, on fungal-colonized leaves, viruliferous aphids were more negatively impacted than virus-free ones and nymph mortality was significantly higher than on fungal-free plants. Furthermore, we assessed PLRV transmission by *M. persicae* on tobacco plants inoculated with either *B. bassiana* or *Metarhizium acridum* ((Driver and Milner) JF Bischoff, Rehner, and Humber) as source or/and recipient plants. *Myzus persicae* was found to acquire and transmit PLRV regardless of the treatment. Nevertheless, the infection rate of endophytically colonized plants was lower at a seven-day incubation period and had increased to almost 100% after fifteen days. These results suggest that *B. bassiana* is effective against aphids, both by contact and via endophytism, and both *B. bassiana* and *M. acridum* delayed PLRV infection in tobacco.

## 1. Introduction

Aphids are major crop pests, not only due to their feeding on phloem sap but mainly because they are plant virus vectors that induce high yield decreases, from 20 to 70% in potato crops [1,2]. Aphids transmit more than half of all insect-vectored plant viruses in different modes [3]. Firstly, plant viruses are acquired and transmitted by non- and semi-persistent modes by successive brief stylet punctures in epidermal and mesophyll cells. The viral particles are essentially retained on the stylet tip of vectors, the latter becoming immediately viruliferous for a short duration (from seconds to minutes) [4]. Secondly, a persistent transmission mode requires a longer phloem-feeding duration by vectors for virus acquisition/transmission. The latent period needs to be higher to allow for virus circulation into the vector before it becomes viruliferous [5,6,7]. For example, Potato leafroll virus (PLRV, Luteoviridae, *Polerovirus*) is most efficiently transmitted by *Myzus persicae* (Sulzer) in a persistent circulative mode [7,8,9]. To achieve an efficient transmission rate, more than 48 h of acquisition and inoculation access periods (AAP and IAP, respectively) and from 8 to 123 h of latent periods are needed [7,10,11]. However, regardless of the transmission mode, plant viruses need to multiply and move within different plant organs prior to symptom appearance and probably to be acquired by the vector during phloem feeding.

To control the spread of persistently transmitted viruses, especially PLRV, the use of synthetic chemicals against vector populations was considered to be the best solution for a long time [5,12,13,14]. Indeed, vector mortality probably occurs before the end of the virus transmission process, particularly during the latent period [11]. However, due to environmental side-effects, mainly on non-target beneficial organisms, several chemicals are currently prohibited to be used in agriculture [15,16]. Consequently, alternatives methods with less ecological impact are encouraged [17].

Entomopathogenic fungi (EPF) are biological control agents as alternatives to synthetic chemicals to control sap-feeding pests such as aphids [18,19,20,21,22,23,24,25,26]. To improve the effectiveness of this biocontrol agent, researchers are focusing on the relationships between EPF and insects through their shared host plant [21,27,28,29,30,31,32]. Among EPF, *Beauveria* spp. (Vuill.) and *Metarhizium* spp. (Sorokin) (Ascomycota: Clavicipitaceae) are two genera of filamentous fungi that are used as the most commercialized fungal biopesticides [33,34,35,36,37]. They are commonly used in inundative treatments via foliar application [38,39,40,41]. Their capability to transcutaneously infect insects is their major asset [39]. Moreover, recent studies have highlighted their ability to be assimilated and to live internally in plant tissues (i.e., endophytically) without any symptom appearance [42,43,44,45,46,47]. Then, they directly interact with plant pathogens and pests, including virus vectors [48,49,50,51].

The effectiveness of several strains of EPF applied by contact has been demonstrated [20,25,52]. However, insect cadavers showing EPF outgrowth as evidence that mortality was caused by fungal infection were sometimes omitted [53] or very low [52,54,55,56]. The latter may have been due to environmental conditions such as temperature and humidity [57,58,59] but also because insect mortality could be due to the indirect effects EPF through host plants. In fact, studies have shown that endophytic entomopathogenic fungi (EEPF) improve plant resistance by increasing the biosynthesis of secondary metabolites in plant tissues that are toxic for insects [32,60,61] and by activating an induced systemic resistance (ISR) in the plant [21,61,62], which can lead to insect mortality. Then, a direct effect of EPF by contact and an indirect effect via endophytism would act synergistically to improve plant fitness. Furthermore, changes in the behavior of virus vectors in response to an EEPF-colonized plant, as well as their infectious status, were reported on host-seeking and plant-feeding behaviors. Indeed, on the one hand, *M. persicae* discriminated against tomato plants (*Solanum lycopersicum* L.) colonized with *B. bassiana* [63] and switched from PLRV-infected plants to healthy potato plants (*Solanum tuberosum* L.) after virus acquisition [64]. On the other hand, significant changes were detected on electrical penetration graph (EPG) variables, including the intracellular probe (pd) duration and other sequential variables, when *Aphis gossypii* (Glover) fed on *B. bassiana*-colonized melon plants (*Cucumis melo* L.) [50] and *M. persicae* fed on Potato virus Y (PVY)-infected tobacco [65]. Additionally, *M. persicae* carrying PLRV or not were found to be both attracted to the tobacco leaves colonized by *Beauveria bassiana* ((Balsamo-Crivelli) Vuillemin) or *Metarhizium acridum* ((Driver and Milner) JF Bischoff, Rehner, and Humber) [66]. This was not the desired effect because biological control agents are supposed to protect plants from pests [63,66]. Therefore, estimating the fitness of an insect living on an EEPF-colonized plant and its ability to transmit a phytovirus is decisive for assessing the true role of EPF endophytism. In addition, it was reported that the susceptibility of viruliferous compared to non-viruliferous insects may differ [67].

In this study, we first evaluated the virulence effect of *B. bassiana* applied directly by contact or indirectly via endophytism through tobacco plants against *M. persicae* carrying PLRV or not. Secondly, we assessed the transmission rate of PLRV by *M. persicae* on tobacco plants inoculated with either *B. bassiana* or *M. acridum* as recipient plants and fungal sources, respectively.

## 2. Materials and Methods

### 2.1. Plants, Insects and Virus

Tobacco seeds (*N. tabacum* cv. *Xanthii*) were placed in a germination container with autoclaved potting soil. Seedlings at the three-leaf stage were individually transferred into pots (7 × 7 × 7 cm^3^) and stored in a growth chamber at 22 ± 1 °C, with 70 ± 10% relative humidity (RH) and a 16 h light period.

A colony of an MpCh4 strain of *M. persicae* was maintained on tobacco plants in 60 × 60 × 60 cm^3^ net cages (Bugdorms, MegaView Science Co., Taichung City, Taiwan) and placed in air-conditioned rooms as described above. Plants were replaced every 3 weeks by four-leaf stage seedlings.

Potato leafroll virus was acquired from infected *Physalis floridana* Rydb. from DSMZ (Deutsche Sammlung von Mikroorganismen und Zellkulturen GmbH, Braunschweig, Germany). It was then transferred on tobacco plants by releasing five *M. persicae* previously virosed on *P. floridana* for 4 days. Similarly, PLRV was maintained by transferring insects from virus-infected tobacco to new healthy seedlings every 3–4 weeks.

### 2.2. Entomopathogenic Fungi

The GHA strain of *Beauveria bassiana* and the IMI330189 strain of *M. acridum* were isolated from the wettable powders of Botanigard and Green Muscle commercial bioinsecticides, respectively. Thirty-five microliters of the respective product suspended in sterile distilled water with 0.01% Tween^®^ 80 were spread on potato dextrose agar (PDA) with additional chloramphenicol (0.05 g/L) to prevent bacterial contamination. Sealed plates were then placed in a light-free incubator at 25 ± 1 °C for 3 weeks. Spores were harvested by scraping the surface of the agar with a sterile L-shaped spreader and suspended in sterile distilled water with 0.01% Tween^®^ 80. The initial concentration of the spore suspensions was determined using a Neubauer hemocytometer cell. The final suspensions were adjusted to 10^8^ spores/mL and stored at 4 °C. Spore viability was assessed on ready-to-use suspensions by spreading three replicates of 0.1 mL of a diluted suspension at 10^4^ spores/mL on a microscopic slide overlaid with a thin slice of PDA. The germination rate was estimated by counting 100 spores using a 200× magnification microscope after incubation for 24 h at 25 ± 1 °C in the dark. Only suspensions with >96% viability were used within a maximum of 24 h.

### 2.3. Aphid Fitness Bioassays

Both plants (inoculated with *B. bassiana* or not) and insects (sprayed with the *B. bassiana* spore suspension or not) were used with viruliferous and non-viruliferous aphids. Fourteen combinations were tested, as described in Table 1. *Metarhizium acridum* was tested in this experiment but not accounted for any further due to the low occurrence of endophytically colonized leaves and their distribution among concerned treatments (see Results section).

Tobacco plants at the four-leaf stage were inoculated with *B. bassiana* (n = 6 per treatment) 48 h before the beginning of the experiment. Two basal leaves were sprayed with 2 mL of a *B. bassiana* suspension using a cosmetic hand spray with a fine mist (0.35 mm nozzle diameter). A plastic zip bag was used to isolate the plant top part in order to prevent any contact with the spores. For fungal-free plants, the fungus suspension was replaced by sterile water with Tween 80 (0.01%). All plants were then covered with a plastic cover (Natureflex™, 160 × 300 mm^2^) and kept in a climate chamber at 22 ± 1 °C and a 16 h light period.

Aphids that were used in this bioassay were obtained by placing 40–50 adults on five healthy or PLRV-infected plants for 24 h. Then, adults were removed, and nymphs of the same age were maintained on the plants for four days, corresponding to the virus acquisition access period (AAP). Groups of 5 individuals from healthy (n = 48) and PLRV-infected (n = 48) plants were then separately placed in Sterilin^®^ Petri dishes (VWR, Radnor, PA, USA) measuring 35 mm in diameter. Both groups from viruliferous and virus-free aphids were also divided into two groups. First, 24 from each group were sprayed with 1 mL of *B. bassiana*, while the remaining other 24 were sprayed with 1 mL of sterile distilled water containing 0.01% Tween 80. Finally, 5–10 min after treatment, each 24 aphid batch was further divided into two final groups of twelve. Each of these final sets of twelve aphids was transferred to either a fungal-free plant or a *B. basiana*-inoculated plant. Five individuals were placed on the two unsprayed leaves of each plant, confined in a clip cage (15 mm diameter and 9.0 mm thickness) [68] made of a polyethylene sleeve for an air-conditioner insulation system (NMC International SA, Weiswampach, Luxembourg).

Every two days for eight days, aphid mortality and fecundity were recorded. Newly deposited nymphs were counted and maintained in clip cages. Aphid fecundity was calculated by dividing the number of newly deposited nymphs by the adult number previously recorded. Adult and nymph cadavers were counted and removed from clip cages. Adult cadavers were transferred into Petri dishes (9 cm diameter) with moist filter paper and incubated at 25 °C for fungal outgrowth examination [52]. At the end of the experiment, all leaves used for the test were immediately sampled and assessed for EEPF colonization. The entire experiment was repeated twice.

### 2.4. Virus Spread Bioassay

Fifteen plants at the three-leaf stage were infected by confining 5 individuals from PLRV-infected plants in a clip-cage for 5 days. They were then evaluated after 14 days of incubation by the DAS-ELISA (double-antibody sandwich-enzyme-linked immunosorbent assay) (see below). Nine plants with the highest optical density (OD) were selected as virosed source plants. They were separated into 3 groups of three plants. Two groups were inoculated with EPF following the process described above. The first group was inoculated with *B. bassiana*, the second was inoculated with *M. acridum,* and the third was sprayed with sterile water with 0.01% Tween 80. Five days post fungal inoculation (dpi), 3 samples from 3 non-sprayed leaves were collected for EEPF examination. At 8 dpi, several adult individuals were released on the source plants (SPs) in order to reproduce. Twenty-four hours later, adults were removed and laid nymphs were maintained for 4 days, corresponding to the virus AAP. Following the EEPF colonization rate, only 1/3 of SPs for the same treatment with the highest EEPF colonization rate (for treated plants) were selected for the bioassays.

Three treatments of recipient plants (RPs) were prepared: *B. bassiana* (Bp)-inoculated plants, *M. acridum* (Mp)-inoculated plants, and fungal-free plants (Ffp) as a control. Those plants were at the four-leaf stage. Their two basal leaves were sprayed with the corresponding inoculum or sterile water with Tween 80 at 0.01%. At 5 dpi, two leaves (one treated and one non-treated) were sampled for the confirmation of the EEPF colonization of tissues. Each treatment of RP was divided into 3 subgroups of 12 plants. Each SP group corresponded to 3 RP subgroups according to the combinations (see Table 2).

At 7 dpi, five individuals from the SP selected for each treatment were confined in a clip cage on one of the RP leaves that were not exposed to the fungal spray. The IAP was 3 days, after which aphids were removed with a brush. Plants were kept at room temperature, as described above for incubation. Samples were collected on the 7th, 11^th^, and 15th days of incubation from the top of the plants where the virus concentration was probably higher [69].

The PLRV infection in each plant was assessed by a qualitative DAS-ELISA with a DSMZ kit following the manufacturer’s instructions in triplicate. Plants were considered as infected when the average OD was at least two times high than negative controls. The entire experiment was repeated thrice.

### 2.5. Confirmation of Endophytic Colonization of Tobacco by EPF

Leaves collected immediately after the fitness bioassays, and at the 5th dpi for virus spread, bioassays were evaluated to confirm EEPF colonization. They were first surface-sterilized with a solution containing 0.5% NaOCl and Tween 80 (0.01%), then sterilized in a solution containing 70% ethanol, and finally rinsed thrice in sterile distilled water. After drying on sterilized paper towels, 6 samples of approximately 1.5 cm^2^ from each leaf were taken using a sterile scalpel blade. Samples from each leaf were first pressed (adaxial and abaxial sides) on a PDA plate to mark an imprint to determine whether any epiphytic spores remained on the leaf surface and next transferred on a new culture medium to incubate. Three samples of 100 µL of final rinse water from each leaf were plated on the PDA to evaluate the disinfection process. All plates were sealed and placed in a light-free incubator at 25 °C for 10 days. Fungal colonies growing from internal plant tissues were visually examined according to the characteristics described by the authors of [70,71] for *B. bassiana* and *M. acridum,* respectively. For insect fitness bioassays, when one tissue from a tested leaf showed fungal growth, the whole leaf was classified as being endophytically colonized [43]. The EEPF colonization rate was calculated for each leaf using Formula (1). In any case, only results from leaves with confirmed EEPF colonization were taken into account.
(1)$$EEPF\;colonisation\;rate=\left(\frac{number\;of\;colonised\;leaf\;tissues}{total\;number\;of\;leaf\;tissues}\right)\times 100$$

### 2.6. Statistical Analysis

Aphid fitness data collected on the eighth day of the experiment were processed using the R software. A multiple linear regression model was fitted to compare treatments with plant type, aphid virosed status, fungal application mode, and interactions between factors. The factor experiment was not significant and was therefore removed from the model. Then, data from two replicates of fitness assays were pooled in the same analysis. Requirements for the validity of the test (normality and homoscedasticity of the residual) were checked before applying the analysis of variance (ANOVA) test to the regression model. For each significant difference, a pairwise comparison was applied using the “lsmeans” function of the lsmeans package [72] with the Tukey fitting method. The “cld” function of the multcomp package [73] was used to set up a compact display of the letters of all pairwise comparisons. In addition, the probability of adult survival was estimated using the Kaplan–Meier method [74], and curves were plotted with “survival” and “survminer” packages [75]. The multivariate Cox proportional hazards [76] were also calculated to analyze the effects of factor combinations on adult survival using same packages.

Correlations between the colonization rate of leaf tissue by EEPF and insect fitness data including mortality, fecundity, and nymph mortality rates were evaluated by the Spearman method using the “cor.test” function of the package stats [77,78].

The proportions of PLRV-infected plants to uninfected plants for each incubation period were assessed by Pearson’s chi-squared test. The pairwise tests of independence for nominal data were applied with the “pairwiseNominalIndependence” and the “cldList” functions of the rcompanion package [79] to determine the symmetry between treatments.

The significance threshold for all tests was set at 0.05%.

## 3. Results

### 3.1. Endophytic Colonization of Tobacco Plants

The inoculation technique of *B. bassiana* and *M. acridum* showed successful colonization rates of leaves ranging from 16.7 to 100.0%. Then, 62 and 12 individual leaves inoculated to *B. bassiana* and *M. acridum,* respectively, were considered as endophytically colonized for insect fitness trials. Due to the low occurrence of endophytically colonized leaves and their distribution over the concerned treatments (Mp:I−:sf = 5, Mp:I−:ms = 2, Mp:I+:sf = 4, and Mp:I+:ms = 0; I+: viruliferous and I−: non-viruliferous), aphid fitness essays with *M. acridum* were not considered for further analysis. However, in virus transmission trials, 83 and 89 recipient plants inoculated with *B. bassiana* and *M. acridum*, respectively, were considered as endophytically colonized based on the detection of fungal outgrowth within any of the leaf sections sampled on that plants.

### 3.2. Direct and Indirect Effect of EPF on Aphid Mortality

Adult mortality was significantly higher on fungal-inoculated plants compared to fungal-free plants (F = 53.45; *p* < 0.001; Figure 1A). Additionally, the mortality rate of aphids that were directly sprayed with *B. bassiana* was significantly higher compared to unsprayed insects (F = 14.45; *p* < 0.001). Similar observations with aphids with or without virus were obtained (F = 1.73; *p* = 0.189). In contrast, there was a significant effect on the interaction between the aphid infection status and plant treatment (F = 7.68; *p* < 0.01). Indeed, on plants colonized with EEPF, mortality in virosed aphids was higher than the virus-free ones. Furthermore, there was a positive correlation between adult mortality and the rate of leaf tissue colonization to EEPF (rho = 0.47; S = 63; *p* < 0.001). However, neither the endophytic colonization and contact treatment interaction nor the interaction between contact treatment and the aphid virosed state produced significant effects on aphid mortality (F = 0.08 and *p* = 0.771 vs. F = 3.59 and *p* = 0.061, respectively).

Direct aphid spraying with a spore suspension was the only significant factor considering the fungal outgrowth on insect cadavers (F = 60.81; *p* < 0.001). Globally, fungal outgrowth was only observed on the aphid cadavers that were sprayed with *B. bassiana* (Figure 1B). However, a few cadavers from treatments with plants colonized by EEPF showed fungal outgrowth, but it was not significantly different from those from fungal-free plants (F = 0.27; *p* = 0.601).

Survival probability according to treatments revealed a significant difference (X^2^ = 85.2; df = 7; *p* < 0.001). Aphid survival probability tended to drop between the second and fourth days of EPF spray treatments (Ffp:I−:bs, Ffp:I+:bs, Bp:I−:bs, and Bp:I+:bs; Figure 2). In contrast, treatments made with EEPF-inoculated plants and EPF-unsprayed insects (Bp:I−:sf and Bp:I+:sf) had an accelerated decrease in survival on the sixth and eighth days of observation. Curves related to fungal-free plants were not significantly different except for the Ffp:I+:sf treatment, which had the highest survival probability (92.5%) (Table 3). However, the Bp:I+:bs treatment was the other extreme (47.5%), the only one having reached the median lethal time (LT50) after eight days. Endophytic EPF-inoculation on plants and EPF-spray on aphids were strongly related to mortality risk. Indeed, the fungal-free plant condition was associated with a lower mortality (hazard ratio: HR = 0.41; *p* < 0.001), while spraying EPF on aphids significantly increased mortality (HR = 1.72; *p* < 0.001).

### 3.3. Direct and Indirect Effect of EPF on Aphid Fecundity and Nymph Mortality

Aphid fecundity analysis per capita was negatively impacted by the presence of EEPF in host plant tissues (F = 228.17; *p* < 0.001; Figure 3A). In addition, a negative correlation between leaf colonization rate by EEPF and the fecundity per capita was observed (rho = −0.66; S = 20; *p* < 0.001). Similarly, in treatments where aphids were exposed to a direct contact with *B. bassiana*, the reproduction rate was significantly decreased (F = 37.36; *p* < 0.001). Different responses were observed according to aphid virosed status and the *B. bassiana* colonization of their host plant. The fecundity rate of virosed aphids was significantly lower than virus-free ones (F = 12.06; *p* < 0.001). Finally, there was no synergistic effect between contact and endophytic plant colonization with EPF (F = 0.45; *p* = 0.501).

The mortality rate of nymphs laid on endophytically-colonized leaves were significantly higher than those laid on non-treated leaves (F = 194.21; *p* < 0.001; Figure 3B). In addition, a strong positive correlation was detected between nymph mortality and the *B. bassiana* colonization rate of leaf tissues (rho = 0.72; S = 331,765; *p* < 0.001). The direct spraying of *B. bassiana* spore suspensions did not significantly affect the nymph mortality (F = 1.59; *p* = 0.208). Similarly, there was no significant effect of insect virosed status (F = 0.003; *p* = 0.957). In contrast, a significant effect of interaction between direct fungal spraying and aphid virosed status was observed (F = 4.57; *p* < 0.03).

### 3.4. Effect of EEPF Colonization on Virus Spread by Aphids

The effect of EEPF on the virus transmission rate was significant for the seven-day (X^2^ = 50.94; df = 8; *p* < 0.001) and eleven-day (Χ^2^ = 40.00; df = 8; *p* < 0.001) incubation periods (Figure 4A,B). The highest contamination rates were observed at the Ffp–Ffp and Ffp–Mp treatments with 94.4% and 83.3% of infected plants, respectively, on the seventh day of the incubation period. The lowest contamination rates on the seventh day of incubation were observed for the Bp–Bp and Bp–Mp treatments with 33.3% of infected plants. This had gradually increased to 41.7% and 66.7% for Bp–Bp and Bp–Mp, respectively, and reached 100% at the 15th day of incubation. An overall reduction in the PLRV incidence was observed on recipient plants treated with *B. bassiana* (Mp–Bp and Ffp–Bp) and/or whose aphid vectors were derived from plants inoculated with *B. bassiana* (Bp–Ffp, Bp–Mp, and Bp–Bp). An intermediate level of contamination was observed on treatments from combinations of fungal-free plants and those inoculated with *M. acridum* as source or recipient plant (Mp–Ffp, Mp–Mp, and Ffp–Mp).

## 4. Discussion

The evaluation of the EEPF inoculation technique revealed successful colonization of tobacco leaves by *B. bassiana* and *M. acridum* from 16.7 to 100.0% of leaf tissues from unsprayed leaves on the same plants as that were treated with a spore suspension. However, there were less *M. acridum*-colonized leaves for insect fitness trials than those for virus propagation. This was justified by the inclusion of leaves sprayed with spore suspensions for virus propagation trials. The improvement of endophytic colonization-detection techniques could help to properly quantify the extent of plant colonization to EEPF by reducing the risk of false negatives [27].

The effectiveness of *B. bassiana* as a biological control agent against phloem-feeding insects and especially aphids are well-known and widely documented [21,26,28,30,80,81,82,83,84,85,86]. The most common treatment method use in the field and in the laboratory is the direct contact between the EPF spores and the insect pest [87,88,89,90]. It is also known that the infection of EPF occurs through spore adhesion and germination on the insect cuticle, penetration into the organism by enzymatic activity and mechanical pressure, the invasion of the insect tissues and diffusion of toxins leading to death, and, finally, mycelia growth on the insect cadaver in the saprophytic phase of EPF [91,92,93]. In our study, we showed that the mortality rate was significantly increased and that there was *B. bassiana* mycelia growth on the aphids that were sprayed with the spore suspension. Additionally, the mortality rate was significantly higher in aphids settled on the plant leaves that were endophytically colonized by *B. bassiana*. These results are in accordance with several studies such as those from Gurulingappa et al. [94] and Lopez et al. [30], who evaluated *A. gossypii* survival on cotton (*Gossypium hirsutum* L.) colonized to *B. bassiana*, as well as Manoussopoulos et al. [26], who used *M. persicae* and *Chaetosiphon fragaefolii* (Cockerell) on strawberry *Fragaria × ananassa* (Duchesne) plants. These studies provided evidence that *B. bassiana* was also indirectly effective against aphids via endophytism. Even if Allegrucci et al. [81] did not find significant aphid mortality differences on pepper (*Capsicum annuum* L.) endophytically colonized by *B. bassiana* and control plants, aphids more numerously died on *B. bassiana*-inoculated plants. Such an increased mortality in aphids settled on endophytically colonized plants was previously attributed to a possible improvement in plant resistance conferred by EEPF [27,95]. This hypothesis was supported by our results that showed a positive correlation between adult mortality and leaf EEPF colonization rate. Unlike aphids sprayed with the EPF spore suspension, the mycosis rate was very low (1–3%) in aphid cadavers derived from endophytically colonized leaves. In most published studies concerning the effect of EEPF on insects, mycosis has either not been reported [96,97] or not observed [94,98], except for in stem-borers or leaf mining insects [31,99]. For phloem-feeding pests including aphids, some studies have suggested that EEPF provide the overall insecticidal protection by the biosynthesis of toxic secondary metabolites that could: (1) directly reach insects via plant tissues in which they feed [32,60,61] or (2) induce a systemic resistance by stimulating the plant immune system [21,62] causing feeding deterrence, antibiosis, or global changes in metabolism of the host plant, thus impacting its suitability towards herbivores [43]. Mycelium can also be found in the epidermal cells, palisade parenchyma, intercellular spaces, and vascular elements of the xylem of EEPF-colonized plants [44,100]. This could possibly infect insects, as observed in our study.

The number of nymphs produced per adult was significantly lower with treatments where the insects were sprayed with the spores or/and their plant was endophytically colonized with *B. bassiana*. Regarding treatment by contact, similar results were obtained by Baverstock et al. [86] with *Acyrthosiphon pisum* (Harris) infected with *Pandora neoaphidis* and *B. bassiana*, as well as by Gurulingappa et al. [32] with *A. gossypii* sprayed with spores of *Lecanicillium lecanii* (Zimm.) and *B. bassiana*. Shrestha et al. [87] also found that daily nymph production and overall fecundity rate in lettuce aphid, *Nasonovia ribisnigri* (Mosley) were significantly affected when treated with *B. bassiana*. During pathogenesis, the sublethal level of EPF could affect the insect’s ability to reproduce through physical damage and intoxication, with secondary metabolites affecting insect reproductive system [87]. This was possibly the mechanism of action responsible of the decrease of aphid fecundity in our study. Furthermore, many other studies have shown that the EEPF colonization of plants significantly reduces aphid fecundity [28,30,32,101]. In addition, Jaber and Araj [28] showed that the fecundity rate of *M. persicae* settled on sweet pepper leaves colonized by *B. bassiana* and *Metarhizium brunneum* (Petch) was significantly reduced over two generations. Interestingly, our results showed a strong negative correlation between the aphid fecundity rate and the EEPF colonization leaf rate. This suggested that the presence of *B. bassiana* decreased *M. persicae* fecundity. Similarly, Collinson et al. [102] reported a significant decrease in fecundity rates for *Diuraphis noxia* (Mordvilko) and *Aploneura lentisci* (Passerini) on perennial ryegrass cv. Alto (*Lolium perenne* L.) endophytically colonized by *Epichloë festucae*. The authors associated these results with the presence of the alkaloids lolitrem B, ergovaline, and peramine identified in colonized plants [102]. *Beauveria bassiana* produces several molecules in vivo and in vitro with antimicrobial, insecticidal, and cytotoxic activities such as bassianin, tennelin, beauvericin, bassianolide, pyridomacrolidin, and pyridovericin [61,103,104,105]. However, future investigations should establish a link between the biosynthesis of alkaloids in planta in sufficient amounts required to affect herbivore fitness and evidence of their impact [61,94].

In our case, it was widely demonstrated that the GHA strain of *B. bassiana* was effective against aphids, including *M. persicae* [80,86,87,106,107]. Moreover, there was a strong positive correlation between the mortality rate of nymphs and the EEPF colonization rate of leaves on which nymphs were deposited. In addition, our findings showed that the spore spraying treatment had no impact on nymph survival. This would indicate that the nymphs would not be infected by the EPF. Therefore, there would have been no spore transmission, either vertically or horizontally to the nymphs [85]. The precautions taken during the experiment, in particular removing dead insects regularly from the clip cage, helped to avoid the nymph contamination.

Presently, few studies have provided information regarding the susceptibility of virus vectors to biological control agents, particularly when the vectors are carrying phytovirus compared to virus-free aphids. For instance, *Rophalosiphum padi* was found to be more susceptible to be parasitized by *Aphidius colemani* (Viereck) when carrying Barley yellow dwarf virus (BYDV) compared to non-viruliferous individuals [108]. Here, our findings revealed that the adult mortality rate recorded with the Bp:I−:Bb treatment was significantly lower compared to its homologous treatment with viruliferous insects (Bp:I+:Bb). In addition, the survival curve of the latter treatment was significantly lower compared to the others. There was also a higher average fecundity for the Bp:I−:Sf treatment than its homologous treatment with viruliferous individuals (Bp:I+:Sf). To our knowledge, we showed for the first time that the virosed state influenced aphid fitness when exposed to an EPF. Virus transmission by aphids leads to complex chemical reactions that mediate the interactions between the plant, the vector, and the virus [109]. Indeed, plants have developed several mechanisms of genetic, metabolic, and physiological resistance that act in complementary ways to promote their defense against viral infections and insect vector infestations [110]. Aphids produce effector proteins contained in their watery saliva that are injected into plants during phloem-feeding with various effects on plant defense according to host species. Within saliva, some viruses are transmitted with induced impact on plant responses. Depending on the kind of virus, the latter can improve plant quality in order to attract non-viruliferous aphids to facilitate viral propagation [111]. A complete analysis of the chemical ecology in this pathosystem would allow us to determine the responses of each component towards EEPF [109]. Such information, as well as the vectorial competence of the insect settled on EEPF-inoculated plants, is very important to understand the plant–EPF–insect–virus relationship, in particular the role of EPF on virus propagation in the field in a multitrophic approach of biological control toward a diversity of bio-aggressors, including pests and diseases.

Regarding the vectorial competence of aphids for an EEPF-colonized plant, overall, *M. persicae* was able to acquire and to transmit PLRV to a new plant regardless of the source and recipient plant treatment. Indeed, nearly all kinds of plants were found to be positive for PLRV by an ELISA after fifteen days of incubation. This was in accordance with the findings of Gonzalez et al. [50], where the EPG variables related to the inoculation of persistently and non-persistently transmitted viruses were not altered when *A. gossypii* was fed on *C. melo* endophytically colonized by *B. bassiana*. Additionally, a similar study reported significantly lower rates of the transmission of Cucumber mosaic virus (CMV) and Cucurbit aphid-borne yellows virus (CABYV) by *A. gossypii* on *B. bassiana*-colonized plants compared to control plants [50]. In our study, the early evaluation revealed a significantly lower infection rate on recipient plants inoculated with EEPF compared to fungal-free plants, especially at the seventh day of incubation. This potential plant resistance against the virus conferred by EEPF was also reported by Jaber and Salem [112], who found that *B. bassiana* significantly reduced the incidence and severity of Zucchini yellow mosaic virus (ZYMV) on squash (*Cucurbita pepo* L.). Similarly, the fungal *Hypocrea lixii* Pat. significantly reduced the disease level caused by Iris yellow spot virus (IYSV) on onion (*Allium cepa* L.) [113].

Various mechanisms may justify the reduction of virus incidence on EEPF-colonized plants. This include the activation of the plant’s defense system and the antibiosis effect of EEPF secondary metabolites against viruses [50,61,112,113,114,115]. Accordingly, oosporein, one of the major substances synthesized by *B. bassiana*, and destruxins synthesized by *Metarhizium* spp. have antiviral activity [60,104,116]. Destruxins have been reported to be produced in endophytically colonized plant tissues [116], but nothing has been reported about oosporein. In all cases, the interaction of destruxins with plant-viruses in plant tissue should still be demonstrated. Furthermore, space and resource competition between EEPF and a virus could directly affect the location and movement of a virus in plant tissues [50,112,117]. Endophytic strains of *B. bassiana* and *Metarhizium* spp. can systemically reach all parts of the plant-colonizing intercellular spaces and vascular xylem elements upwards and downwards from the inoculation point [100,112,118,119]. Then, the intercellular movement of the virus could be delayed or even inhibited by the EEPF [112]. This could possibly explain the low infection rate detected in our study during early plant evaluation. Jaber and Salem [112] found that ZYMV symptom development was delayed after mechanical inoculation on squash colonized with *B. bassiana*. However, the plant protective duration by EEPF against the virus would have been limited. Indeed, the colonization rate of plant tissue decreased with time [81] depending on several factors, including the inoculation method, fungal isolate, plant species, and environmental conditions [36,46,47]. For instance, in tobacco, tissue colonization by *B. bassiana* inoculated by the foliar spray method dropped from 100 to 20% twenty days after inoculation [47]. Then, in our study, the fifteenth day of incubation corresponded to the twenty-third day after fungal inoculation. The increase in the virus infection rate after late plant evaluation would have been the result of EEPF decline in plant tissue.

## 5. Conclusions

This study confirmed the effectiveness of *B. bassiana* as a biological control agent against aphids and especially *M. persicae*. The effects of *B. bassiana* on aphid fitness were observed by using both direct contact by spore spraying and indirect exposure via endophytic setting. There was no interaction between the two modes of application. However, to our knowledge, we report, for the first time, that virosed aphid fitness was negatively impacted compared to virus-free vector when exposed to EPF. In addition, in *B. bassiana*- and *M. acridum*-colonized plants, the infection of PLRV on plants was delayed for about one week compared to the control. This suggests a possible plant resistance against virus due to the presence of EEPF in plant tissues. Analyzing the chemical ecology in this pathosystem would allow us to determine the role of EEPF in order for it to be integrated to a crop protection strategy in multitarget pest.

## Figures and Tables

**Figure 1 insects-12-00089-f001:**
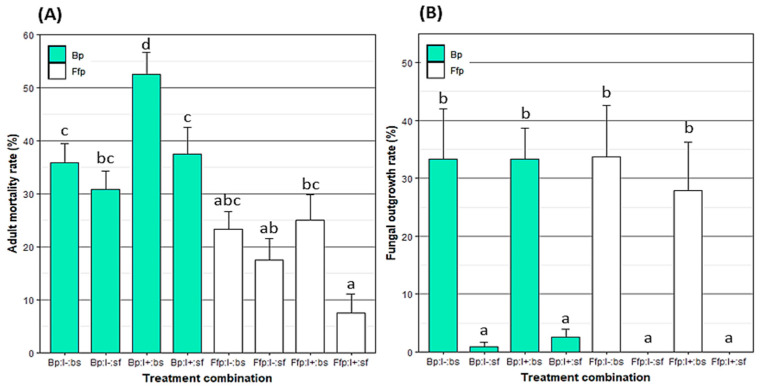
Mean ± SE of (**A**) mortality rate (%) of *Myzus persicae* eight days after exposure to a spray of *B. bassiana* and/or release on *B. bassiana*-inoculated plants and (**B**) fungal outgrowth rate (%) from *M. persicae* cadavers after ten days of incubation. Treatments were based on combinations of plant type (Bp: *B. bassiana*-inoculated plant; Ffp: fungal-free plant), insect infectious status (I+: viruliferous; I-: non-viruliferous) and insect spraying (bs: with *B. bassiana* spore suspension; sf: with spore-free solution). Treatments followed by a different letter differed significantly (*p* < 0.05). n = Ffp:I−:sf, Ffp:I−:bs, Ffp:I+:sf and Ffp:I+:bs = 24; Bp:I−:sf = 16; Bp:I−:bs = 14; Bp:I+:sf = 15; and Bp:I+:bs = 17.

**Figure 2 insects-12-00089-f002:**
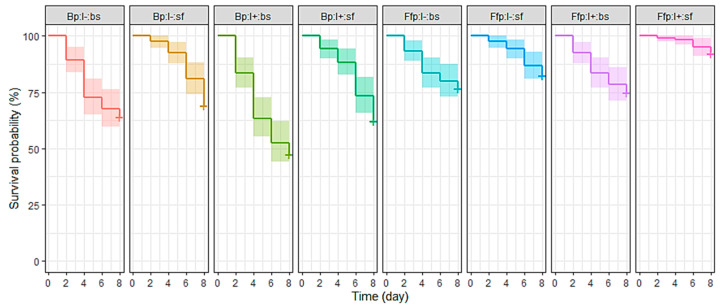
Survival curves of *Myzus persicae* for eight days after exposure to a spray of *B. bassiana* and/or release on *B. bassiana*-inoculated plants. Treatments were based on combinations of plant type (Bp: *B. bassiana*-inoculated plant; Ffp: fungal-free plant), insect infectious status (I+: viruliferous; I−: non-viruliferous) and insect spraying (bs: with *B. bassiana* spore suspension; sf: with spore-free solution). n = Ffp:I−:sf, Ffp:I−:bs, Ffp:I+:sf and Ffp:I+:bs = 120; Bp:I−:sf = 80; Bp:I−:bs = 70; Bp:I+:sf = 75; and Bp:I+:sf = 85.

**Figure 3 insects-12-00089-f003:**
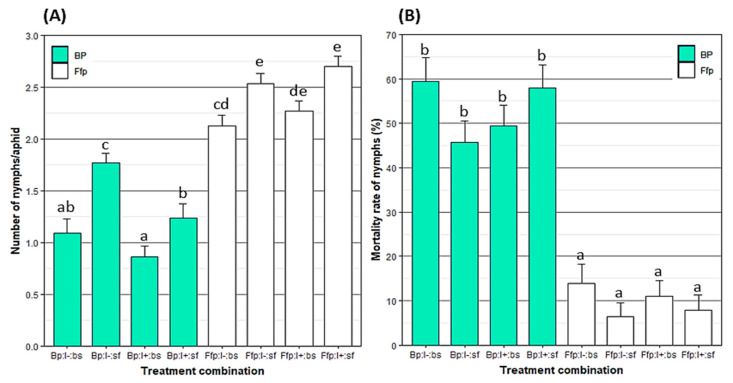
Mean ± SE of (**A**) fecundity per capita of *Myzus persicae* and (**B**) mortality rate of nymphs laid eight days after release on *B. bassiana*-inoculated or fungal-free plants. Treatments were based on combinations of plant type (Bp: *B. bassiana*-inoculated plant; Ffp: fungal-free plant), insect infectious status (I+: viruliferous; I−: non-viruliferous) and insect spraying (bs: with *B. bassiana* spore suspension; sf: with spore-free solution). Treatments followed by a different letter differ significantly (*p* < 0.05). n = Ffp:I−:sf, Ffp:I−:bs, Ffp:I+:sf and Ffp:I+:bs = 36; Bp:I−:sf = 31; Bp:I−:bs = 24; Bp:I+:sf = 22; and Bp:I+:sf = 34.

**Figure 4 insects-12-00089-f004:**
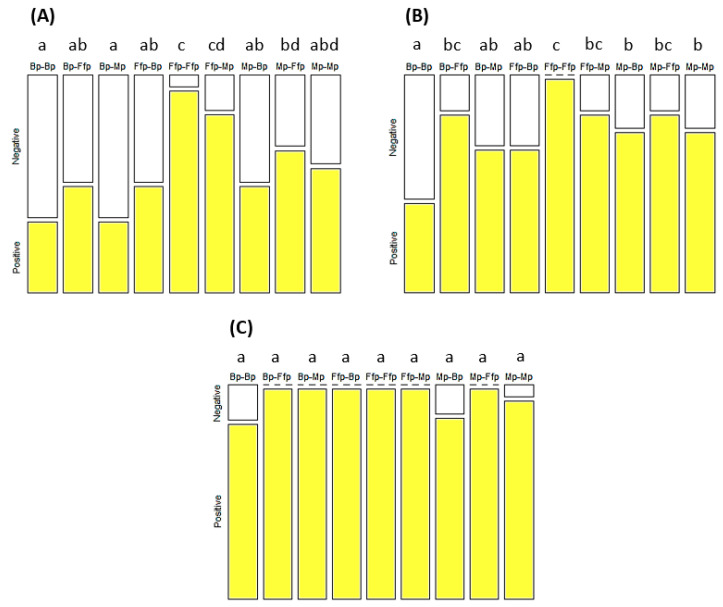
Percentage of PLRV-infected (positive) and non-infected (negative) tobacco after seven (**A**), eleven (**B**), and fifteen (**C**) days of incubation. Treatments were based on combinations of *B. bassiana*-inoculated plants (Bp), *M. acridum*-inoculated plants (Mp), and fungal-free plants (Ffp), as well as a control, as source (two first letters) and recipient (two last letters) plants. Treatments followed by different letters were significantly different (*p* < 0.05). n = Bp–Ffp, Mp–Ffp, and Ffp–Ffp = 36; Bp–Bp = 32; Bp–Mp = 29; Ffp–Bp = 26; Ffp–Mp = 30; Mp–Bp = 25; and Mp–Mp = 30.

**Table 1 insects-12-00089-t001:** Different treatment combinations in the insect fitness bioassays. I+: viruliferous, I−: non-viruliferous.

Plant Treatment	Insect		Treatment	Description
	Infectious Status	EPF Treatment		
Fungal-free (Ffp)	Non-Viruliferous (I−)	Spore-free (sf)	1. Ffp:I−:sf	Non-viruliferous insects, sprayed with spore-free solution, released on a fungal-free plant
		*Beauveria bassiana* spores (bs)	2. Ffp:I−:bs	Non-viruliferous insects, sprayed with *B. bassiana* spore suspension, released on a fungal-free plant
		*Metarhizium acridum* spores (ms)	3. Ffp:I−:ms	Non-viruliferous insects, sprayed with *M. acridum* spore suspension, released on a fungal-free plant
	Viruliferous (I+)	Spore-free (sf)	4. Ffp:I+:sf	Viruliferous insects, sprayed with spore-free solution, released on a fungal-free plant
		*B. bassiana* spores (bs)	5. Ffp:I+:bs	Viruliferous insects, sprayed with *B. bassiana* spore suspension, released on a fungal-free plant
		*M. acridum* spores (ms)	6. Ffp:I+:ms	Viruliferous insects, sprayed with *M. acridum* spore suspension, released on a fungal-free plant
*B. bassiana*-inoculated (Bp)	Non-Viruliferous (I−)	Spore-free (sf)	7. Bp:I−:sf	Non-viruliferous insects, sprayed with spore-free solution, released on a *B. bassiana*-inoculated plant
		*B. bassiana* spores (bs)	8. Bp:I−:bs	Non-viruliferous insects, sprayed with *B. bassiana* spore suspension, released on a *B. basiana*-inoculated plant
	Viruliferous (I+)	Spore-free (sf)	9. Bp:I+:sf	Viruliferous insects, sprayed with spore-free solution, released on a *B. bassiana*-inoculated plant
		*B. bassiana* spores (bs)	10. Bp:I+:bs	Viruliferous insects, sprayed with *B. bassiana* spore suspension, released on a *B. bassiana*-inoculated plant
*M. acridum*-inoculated (Mp)	Non-Viruliferous (I−)	Spore-free (sf)	11. Mp:I−:sf	Non-viruliferous insects, sprayed with spore-free solution, released on a *M. acridum*-inoculated plant
		*M. acridum* spores (ms+)	12. Mp:I−:ms	Non-viruliferous insects, sprayed with *M. acridum* spore suspension, released on a *M. acridum*-inoculated plant
	Viruliferous (I+)	Spore-free (sf)	13. Mp:I+:sf	Viruliferous insects, sprayed with spore-free solution, released on a *M. acridum*-inoculated plant
		*M. acridum* spores (ms)	14. Mp:I+:ms	Viruliferous insects, sprayed with *M. acridum* spore suspension, released on a *M. acridum*-inoculated plant

**Table 2 insects-12-00089-t002:** Different treatment combinations in the virus spread bioassay.

Source Plant	Recipient Plant	Treatment Combination	Description
Fungal-free (Ffp)	1. Fungal-free (Ffp)	1. Ffp–Ffp	Vectors from fungal-free plant released on fungal-free plants (control)
	2. *B. bassiana* plant (Bp)	2. Ffp–Bp	Vectors from fungal-free plant released on *B. bassiana*-inoculated plants
	3. *M. acridum* plant (Mp)	3. Ffp–Mp	Vectors from fungal-free plant released on *M. acridum*-inoculated plants
*B. bassiana* plant (Bp)	4. Fungal-free (Ffp)	4. Bp–Ffp	Vectors from *B. bassiana*-inoculated plant released on fungal-free plants
	5. *B. bassiana* plant (Bp)	5. Bp–Bp	Vectors from *B. bassiana*-inoculated plant released on *B. bassiana*-inoculated plants
	6. *M. acridum* plant (Mp)	6. Bp–Mp	Vectors from *B. bassiana*-inoculated plant released on *M. acridum*-inoculated plants
*M. acridum* plant (Mp)	7. Fungal-free (Ffp)	7. Mp–Ffp	Vectors from *M. acridum*-inoculated plant released on fungal-free plants
	8. *B. bassiana* plant (Bp)	8. Mp–Bp	Vectors from *M. acridum*-inoculated plant released on *B. bassiana*-inoculated plants
	9. *M. acridum* plant (Mp)	9. Mp–Mp	Vectors from *M. acridum*-inoculated plant released on *M. acridum*-inoculated plants

**Table 3 insects-12-00089-t003:** Pair-wise comparison of *Myzus persicae* survival curves for eight days after exposure to a spray of *B. bassiana* and/or release on *B. bassiana*-inoculated plants. Treatments were based on combinations of plant type (Bp: *B. bassiana*-inoculated plant; Ffp: fungal-free plant), insect infectious status (I+: viruliferous; I−: non-viruliferous) and insect spraying (bs: with *B. bassiana* spore suspension; sf: with spore-free solution). n = Ffp:I−:sf, Ffp:I−:bs, Ffp:I+:sf and Ffp:I+:bs = 120; Bp:I−:sf = 80; Bp:I−:bs = 70; Bp:I+:sf = 75; and Bp:I+:sf = 85.

	Bp:I−:sf						
Bp:I−:bs	-	Bp:I−:bs					
Bp:I+:sf	-	-	Bp:I+:sf				
Bp:I+:bs	***	*	**	Bp:I+:bs			
Ffp:I−:sf	*	**	**	****	Ffp:I−:sf		
Ffp:I−:bs	-	+	+	****	-	Ffp:I−:bs	
Ffp:I+:sf	****	****	****	****	*	**	Ffp:I+:sf
Ffp:I+:bs	-	+	-	****	-	-	***

With ****: *p* = 0; ***: *p* < 0.0001; **: *p* < 0.001; *: *p* < 0.01; ‘+’: *p* < 0.05; ‘-’: *p* > 0.05.

## Data Availability

The data presented in this study are available in this article.

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
