# Peer review of "Direct and Indirect Effect via Endophytism of Entomopathogenic Fungi on the Fitness of Myzus persicae and Its Ability to Spread PLRV on Tobacco"

_insects, 2021, doi:10.3390/insects12020089_

Round 1

Reviewer 1 Report

Dear all,

Overall, it was written well, but I do have some suggestion and questions.

  1. Page 1 abstract, line 39 "M. acridum ” first time on the text should be written "Metarhizium acridum"
  2. Page 2, line 81 “EEPF” abbreviation have been used in text but no definition or explanation about it. I assumed it was "Endophytic Entomopathogenic Fungi" Please confirm and correct. 
  3. Page 3, Did PLRV infection confirm by ELISA on tobacco during maintaining of virus population?  What is the transmission rate of virus?  

Author Response

Dear professor,

Thank you very much for your appreciation and suggestions.

Please see the attachment for the point-by-point response to your comments.

Reviewer 2 Report

I reviewed "Direct and Indirect Effect via Endophytism of Entomopathogenic Fungi on the Fitness of Myzus persicae and its Ability to Spread PLRV on Tobacco." I found this paper exceeding difficult to read because of the language problems. There were multiple authors. The person who wrote the discussion, may I suggest, needs to write the manuscript.

The simple summary: Needs complete revision. Instead of writing that aphids cause "huge" yield losses perhaps you can give an estimate of the damage. I wouldn't use the word "huge". The section about chemical weapons- I think this is the wrong word here. I believe you are talking about pesticides. Also pesticides are still widely used in agriculture so please think about this when revising this section.

Abstract:

Line 29-replace "of" with "for"

Line 35-replace "caused higher" with "increased"

Line 36-....and decreased the fecundity rate as compared to the controls.

Line 51- please define non-persistent, semi-persistent, and persistent modes for the reader

Suggest revising intro going from the big picture to smaller increments until you say what you are planning on doing. Language is awkward in this section

Line 64-replace "molecules" with "chemicals"

Line 62-why is the aphid killed? By the virus?

Line 66-replace "increasingly developing" with "used as"

Lines 81 and throughout the text- you frequently refer to EEPF but only EPF is defined

Line 87-can you tell describe how host-seeking and feeding behavior is altered?

Line 96-Your experimental plan is unclear. The effectiveness of B. bassiana to do what? To kill the aphid? Please revise this

Materials and Methods

Line 104-climatic room-is this a growth chamber? Greenhouse?

Line 109-define DSMZ

Line 129-unclear sentence-please revise

Line 130-133: very confusing sentences- please revise

Tables 1 and 2- Please use abbreviations so it makes sense as to what you are doing

Line 249-significantly killed?

Figure 1-Please add what the genus and species of the aphid you used to the figure description

Line 433-infection

Discussion-the discussion is well written

Line 479-twenty

References: Please check formatting. For many of the journal articles you have the titles capitalized

Reference #41- has a lot of question marks

Reference #95-italics for genus and species

Ref #116-Missing info

Author Response

Dear professor,

Thank you very much for your appreciation and suggestions. Please see the response as following:

I reviewed "Direct and Indirect Effect via Endophytism of Entomopathogenic Fungi on the Fitness of Myzus persicae and its Ability to Spread PLRV on Tobacco." I found this paper exceeding difficult to read because of the language problems. There were multiple authors. The person who wrote the discussion, may I suggest, needs to write the manuscript.

R/ Thank you for your remark. However, the paper was written entirely by one person. We are aware that the problem of language is real. Indeed, in our cover letter, we promised to ask for a professional review of the entire manuscript to improve the english grammar and spelling, which is done. We hope you find the current version much better than the previous one. 

The simple summary: Needs complete revision. Instead of writing that aphids cause "huge" yield losses perhaps you can give an estimate of the damage. I wouldn't use the word "huge". The section about chemical weapons- I think this is the wrong word here. I believe you are talking about pesticides. Also pesticides are still widely used in agriculture so please think about this when revising this section.

R/ The simple summary has been completely revised in the light of your comments, in particular with regard to the use of "huge" and "chemical weapons".

Abstract:

Line 29-replace "of" with "for"

R/ "of" has been replaced by "for".

Line 35-replace "caused higher" with "increased"

R/ "caused higher" has been replaced by "increased".

Line 36-....and decreased the fecundity rate as compared to the controls.

R/ The part of the sentence on this line has been rephrased taking into account your remarks.

Line 51- please define non-persistent, semi-persistent, and persistent modes for the reader

R/ we considered that the modes of virus transmission by vector should not be a problem for readers. Following your advice, we have briefly defined each mode of virus transmission by vector.

Line 64-replace "molecules" with "chemicals"

R/ "chemicals" was used instead of "molecules"

Line 62-why is the aphid killed? By the virus?

R/ The sentence highlights the fact that by using synthetic chemicals to control the virus vectors, the latter are “killed” or their mortality occurs as a result of the phytosanitary treatment probably before the end of the inoculation process of the persistently transmitted viruses, since those viruses require more time to be effectively transmitted. The sentence is reworded to be clearer and precise.

Line 66-replace "increasingly developing" with "used as"

R/ We replaced "increasingly developing" with "used as"

Lines 81 and throughout the text- you frequently refer to EEPF but only EPF is defined

R/ Effectively, we had omitted to define "EEPF". This is now done on line 91 of the current version.

Line 87-can you tell describe how host-seeking and feeding behavior is altered?

R/ Examples of alteration of host-seeking and feeding behaviour have been added.

Line 96-Your experimental plan is unclear. The effectiveness of B. bassiana to do what? To kill the aphid? Please revise this

R/ The paragraph of the experimental plan concluding the introduction has been reworded and improved.

Materials and Methods

Line 104-climatic room-is this a growth chamber? Greenhouse?

R/ Climatic room is a "growth room". We specified it in the manuscript.

Line 109-define DSMZ

R/ DSMZ is the German company "Deutsche Sammlung von Mikroorganismen und Zellkulturen GmbH". We defined it in the manuscript.

Line 129-unclear sentence-please revise

R/ The sentence is rephrased.

Line 130-133: very confusing sentences- please revise

R/ The sentence is rephrased.

Tables 1 and 2- Please use abbreviations so it makes sense as to what you are doing

R/ We renamed and adopted abbreviations of each treatment's name according to your recommendations, except for the insect infectious status (I+/I-). We maintained this for consistency with our previous study on the same subject and published in the same journal.

Line 249-significantly killed?

R/ The sentence is rephrased

Figure 1-Please add what the genus and species of the aphid you used to the figure description

R/ The genus and species of aphid used in this study is added.

Line 433-infection

R/ We replaced "infectious status" with "infection"

Discussion-the discussion is well written

Line 479-twenty

R/ Corrected

References: Please check formatting. For many of the journal articles you have the titles capitalized

R/ We checked all references and corrected where necessary.

Reference #41- has a lot of question marks

R/ Corrected

Reference #95-italics for genus and species

R/ Corrected

Ref #116-Missing info

R/ The reference is completed.